# Chemokine-Decorated Nanoparticles Target Specific Subpopulations of Primary Blood Mononuclear Leukocytes

**DOI:** 10.3390/nano12203560

**Published:** 2022-10-11

**Authors:** Anissa Pisani, Roberto Donno, Giulio Valenti, Pier Paolo Pompa, Nicola Tirelli, Giuseppe Bardi

**Affiliations:** 1Nanobiointeractions & Nanodiagnostics, Istituto Italiano di Tecnologia, Via Morego 30, 16163 Genova, Italy; 2Department of Chemistry and Industrial Chemistry, University of Genova, Via Dodecaneso 31, 16146 Genova, Italy; 3Laboratory of Polymers and Biomaterials, Istituto Italiano di Tecnologia, 16163 Genova, Italy

**Keywords:** nanoparticles, chemokine, targeting, PBMCs

## Abstract

Specific cell targeting to deliver nanoparticles can be achieved by tailored modifications of the material surface with chemical moieties. The selection of the cell targets can be optimized by covering the nanoparticle with molecules, the receptor expression of which is restricted to particular cell subsets. Chemokines perform their biological action through 7-TM G_i_-protein-coupled receptors differently expressed in all tissues. We decorated the surface of biocompatible polymer nanoparticles with full-length CCL5, an inflammatory chemokine that attracts leukocytes by binding CCR5, which is highly expressed in blood-circulating monocytes. Our observations showed that CCL5 functionalization does not affect the nanoparticle biocompatibility. Notably, CCL5 NPs delivered to PBMCs are selectively internalized by CCR5^+^ monocytes but not by CCR5^-^ lymphocytes. The efficacy of PBMC subpopulation targeting by chemokine-decorated nanoparticles establishes an easy-to-use functionalization for specific leukocyte delivery.

## 1. Introduction

Nanoparticle (NP) decoration with appropriate active groups is a common approach which endows them with a capacity for biological recognition, and, in principle, this capacity can be used in therapeutic or diagnostic targeting [1].

NP coating with polyethylene glycol (PEG) is a frequently applied method to reduce particle aggregation, prolong circulation time and eventually improve delivery efficiency [2]. Although PEGylation is advantageous, and its anti-aggregation feature is currently used in many pharmaceutical preparations, this modification alone is not sufficient to selectively discriminate target cells. Moreover, growing evidence of adverse immune reactions to PEG molecules is limiting its efficacy [3,4]. The addition of further chemical moieties on the NP surface to exploit receptor-mediated active internalization and reduce the adsorption of unexpected molecules is often required [5,6].

In this framework, functionalization with chemokines is a potential avenue for recognizing/binding to leukocytes in a phenotypically selective fashion depending on chemokine receptor expression. Chemokine receptors are pharmacologically targeted by non-chemokine small-peptide or small-molecule antagonists, especially in HIV treatment [7,8]. These chemicals have been also used to decorate NPs for drug delivery in CXCR4^+^ cancer cells [9,10,11]. This strategy can be restricted if aimed at other chemokine receptors due to the limited number of antagonists and their potential side effects. Chemokine receptors, however, are attracting more and more attention as therapeutic targets. Very recent works using protein NPs incorporating the T22 peptide [12] to deliver diphtheria toxin, inhibiting invasion and metastasis of CXCR4^+^ carcinoma mouse cells [13], or CXCR4-targeted modified PAMAM dendrimers aimed at glioblastoma cells [14] demonstrated the importance of chemokine receptors in selective NP delivery.

Our group previously used full-length CXC chemokines to functionalize inorganic [15] or organic [16] NPs to target cell lines bearing cognate chemokine receptors on their surfaces. The choice to modify the NP surface with an entire chemokine rather than a short peptide or a small antagonist molecule relies on two advantages, underlining the novelty of our approach. A full-length chemokine is supposed to unambiguously bind its cognate receptor at physiological concentrations.

Here, we show that pro-inflammatory, CCL5-decorated NPs can discriminate among primary mixed leukocyte subpopulations differently expressing CCR5. Of note, these data are obtained from media containing human serum, which ensures that any protein corona are representative of those occurring in the human biological milieu. Our results provide information regarding the feasibility of using chemokine-decorated NPs as a reliable method for specific peripheral blood mononuclear cell (PBMC) subset targeting.

## 2. Materials and Methods

### 2.1. Nanoparticle Synthesis and Functionalization

PLGA/Pluronic NPs were prepared and decorated with biotinylated moieties, as previously described [16], with some differences explained in Appendix A.

### 2.2. Nanoparticle Characterization

Control and CCL5 NP polydispersity (PDI), hydrodynamic size and Z potential were measured in batch mode (DLS not used as AF4 detector) at a temperature of 25 °C. Measures were acquired with a Möbiuz instrument (Wyatt Technology, Santa Barbara, CA, USA) equipped with a laser at 532 nm and a scattering angle of 163.5°. DLS acquisition time was set at 1 s and number at 5. Dynals algorithm was employed to analyze correlation functions. The electrophoretic mobility was measured for 5 s at a voltage amplitude of 3 V and 10 Hz electric field frequency. These data were converted into Z potential using the Smoluchowski equation.

### 2.3. Cell Cultures

THP-1 cells (ATCC, Manassas, VA, USA) were cultured as reported in [8] and supplemented with 5% human serum (HS) (Thermo Fisher Scientific, Waltham, MA, USA). Human PBMCs (Lonza Group Ltd., Basel, Switzerland) were grown in RPMI 1640 (Thermo Fisher Scientific, Waltham, MA, USA) supplemented with 5% HS (Thermo Fisher Scientific, Waltham, MA, USA) and 1% penicillin–streptomycin (Sigma-Aldrich, Saint Louis, MO, USA) in a 5% CO2 humidified atmosphere at 37 °C.

### 2.4. Cytotoxicity Assays

Forward scattering (FSC) and side scattering (SSC) were measured by flow cytometry with BD FACSDiva software 6.0 provided by BD Biosciences (San Jose, CA, USA), gating living cells based on FSC and SSC. In total, 5 × 10^4^ events per sample were acquired.

Cell metabolic activity was determined using an MTS (3-(4,5-dimethylthiazol-2-yl)-5-(3-carboxymethoxyphenyl)-2-(4-sulfophenyl)-2H-tetrazolium) (Promega Corporation, WI, USA) assay following the manufacturer’s instructions. Briefly, after 24 h incubation with 65 µg/mL of CCL5 NPs or control NPs, cells were washed twice, resuspended in culture medium and seeded at a density of 5 × 10^4^ cells/100 µL in 96-well plates (Corning, Corning, NY, USA). A 10 µL measure of CellTiter 96 aqueous solution was added to each well, and the plates were incubated for 3 h at 37 °C in a 5% CO2 humidified atmosphere. The orange MTS formazan product was measured on a Sparck multimode microplate reader (Tecan, Männedorf, Switzerland) at a wavelength of 490 nm. For 24 h, 10% DMSO was used as positive control.

### 2.5. Cytokine Release

Cultured THP-1 cells were collected and resuspended at 2 × 10^5^ cells/mL in 0.5% HS/medium at 37 °C. After 30 min at rest, the cells were treated with 65 μg/mL of both CCL5 NPs and control NPs for 2 h, 6 h and 24 h. After the incubation, cells were centrifuged at 300 x *g* for 5 min, and supernatants were collected; the concentrations of cytokines were evaluated in the supernatants with a Luminex MAGPIX Multiplex Reader (Merck, Darmstadt, Germany) according to the manufacturer’s instructions. At each time point, 100 ng/mL lipopolysaccharide (LPS)-induced cytokine release was used as positive control.

### 2.6. NPs Internalization Assays

THP-1 cells (2 × 10^5^ cells/mL) or PBMCs were collected, resuspended in RPMI 1640 (Thermo Fisher Scientific, Waltham, MA, USA) supplemented with 5% HS/medium (Thermo Fisher Scientific, Waltham, MA, USA) and incubated for 30 min at 37 °C for starvation. After starvation, cells were incubated for 45 min at 37 °C with 65 µg/mL NPs (decorated with different concentrations from 0.03 to 0.30 nmol of CCL5 per ng NPs where indicated in the text) and washed twice at 4 °C. THP-1 cells were also stained with fluorescently labeled antibodies Vio^®^ Bright B515 anti-human CCR5 (Miltenyi Biotec, Bergish, Germany), Vio^®^ Bright B515 REA Control Antibody (S) and human IgG1 (Miltenyi Biotec, Bergish, Germany) at the manufacturer’s recommended concentration for 30 min in the dark on ice, then washed twice and resuspended at 4 °C in phenol red-free RPMI 1640 (Thermo Fisher Scientific, Waltham, MA, USA). Cell fluorescence was acquired on gated 5 × 10^4^ events per sample and analyzed by flow cytometry with BD FACSDiva software 6.0 (BD Biosciences San Jose, CA, USA).

### 2.7. Statistical Analysis

GraphPad Prism 8 software was used for statistical analysis (San Diego, CA, USA). Unless differently indicated, data are expressed as mean ± standard deviation (SD). *p*-values were calculated using either AVOVA multiple comparison analysis vs. control columns followed by Dunnett post hoc test. ** *p* < 0.01, *** *p* < 0.001, **** *p* < 0.0001 or unpaired *t*-test with Welch’s correction where indicated.

## 3. Results and Discussion

### 3.1. Nanoparticle Synthesis and Functionalization

PLGA/Pluronic NPs were prepared in a microfluidic-assisted nanoprecipitation of PLGA in the presence of Pluronic F127, which remained on the nanoparticles surface, ensuring its PEGylation (Figure 1).

The NPs were characterized as previously described [17] and had a size around 90 nm (Figure 2) with a zeta potential of −20 mV. By using a biotinylated Pluronic [16], it was possible to decorate the nanoparticles with streptavidin (STP); this process did not significantly change the NP size, but the presence of a mildly positively charged protein switched their zeta potential to positive values (+10 mV). STP binding sites were only partially occupied by the NP surface biotins, allowing those still available to bind other biotinylated molecules in a second functionalization step. The use of a biotinylated CCL5 chemokine did not dramatically alter the average NP size, the breadth of its distribution (thereby discounting aggregation) or surface charge in relation to the STP-covered particles (zeta potential from average +10 mV to +14 mV).

### 3.2. Nanoparticle Biocompatibility

To evaluate the potential toxicity of the particles, we used different approaches. Firstly, we compared the relative size of the THP-1 cells by flow cytometry in the absence or presence of different STP NP (from now on referred to as “NPs”) concentrations (Figure 2, left histogram). This test was performed to exclude NP concentration limits affecting cell viability not related to the surface functionalization. We found that NP concentrations above 65 µg/mL significantly reduced the cell size, as shown by the average FSC measurements represented in Figure 3 and Appendix A. Then, an MTS viability assay was used to evaluate THP-1 viability after administration of 65 µg/mL of NPs functionalized or not with CCL5 (Figure 2, right histogram). No cytotoxicity was induced by NPs or CCL5-functionalized NPs. As-synthesized NPs also did not affect cell viability, and their results are shown in Appendix A.

Furthermore, immune compatibility of the NPs +/− CCL5 was confirmed by the lack of induction of inflammatory cytokine production (Figure 3C). Evaluation of NP immune compatibility is a necessary test if the cell targets are leukocytes, as non-cytotoxic compounds can still provoke unexpected inflammatory reactions later leading to noxious effects [18].

### 3.3. CCL5 Nanoparticle Internalization through CCR5

THP-1 cells were treated with 65 µg/mL of CCL5 NPs or control NPs equally labeled with Atto 610 for 45 min at 4 °C and 37 °C (Figure 3). After incubation, cells were stained with anti-CCR5 FITC antibodies, and the fluorescence was measured by flow cytometry. Figure 4A shows superimposed dot plots of CCR5-stained THP-1 cells separately treated with NPs (red) or CCL5 NPs (blue). In both panels, CCL5 NP fluorescence is higher than that of the control NPs, demonstrating higher binding of CCL5-functionalized NPs to the CCR5-expressing monocytic cell line. At 4 °C, CCR5 fluorescence was high in the presence of both the particles (upper panel), whereas, at 37 °C, we detected a significant decrease in FITC fluorescent median intensity, demonstrating CCR5 internalization. On the other hand, control NPs did not induce the same green fluorescent reduction, although a slight average decrease was present, likely due to unspecific adhesion of the NP to the membrane proteins of the cells, as quantified in Figure 4B. These results demonstrate specific binding and internalization of CCR5 by CCL5 NPs in complete medium despite the presence of protein corona.

To further prove CCR5 internalization dependence on CCL5 functionalization, we reduced 65 µg/mL CCL5 NP uptake by competition experiments with free CCL5 (Appendix A) and increased the amount of chemokine on the NP surface and repeated the internalization experiments (Figure 4C). Increasing CCL5 concentration on NPs by ten times, CCR5 internalization was augmented by roughly five times, as shown in the lower panel of Figure 4B. We did not further explore a higher amount of chemokine because of different limiting factors. Above a certain amount of CCL5, functionalized NPs displayed non-specific adhesion, as may be seen in Figure 3B, at 4 °C, probably due to stochastic protein–protein interaction between the chemokine and the membrane proteins. Moreover, CCR5 binding is saturated by the limited amount on the cell surface.

Though non-functionalized polymer NPs can also adhere to the cell membrane, the receptor-mediated uptake assures greatly increased internalization of chemokine-decorated ones.

### 3.4. CCL5 Nanoparticle Specific/Selective? Internalization in CCR5^high^ Primary Human Monocytes

Targeting specific cell subpopulations is a key feature to develop in nanocarriers for several purposes, including drug delivery or diagnostic goals. To evaluate the selective CCR5-mediated uptake of our NPs, we performed internalization experiments with human primary PBMCs (Figure 5). In this mixed, mononuclear leukocytes population, most of the monocytes expressed CCR5, whereas almost all lymphocytes were CCR5 negative (Figure 5A and Appendix A). In fact, this receptor was mainly involved in monocyte migration towards sites of inflammation. Following Atto 610 particle administration to the whole PBMC culture supplemented with 5% HS, only the monocytes’ MFI rose, whereas the lymphocytes’ fluorescence did not significantly change. Crucially, the ratio between administration of CCL5 NPs and control NPs increased three-fold, demonstrating the enhanced binding of the chemokine-modified particles. Unspecific NP adhesion was also present in monocytes, as expected, due to the plasma membrane properties of this phagocytic leukocyte population. Nevertheless, the significantly higher CCL5 NP binding in the CCR5^+^ subset proved that chemokine functionalization can be used to target a specific cell population of interest expressing the cognate receptor. As previously shown (Figure 4), at 37 °C, CCL5 NPs were quickly taken up, underlining the role of the chemokine decoration of the NP surface.

The possibility to choose different chemokines to select specific target cells makes this method of functionalization very versatile. Chemokine receptors are expressed by all cells in a different manner in every tissue [19]. By choosing the proper biotinylated chemokine, we could theoretically prepare nanovectors targeting only one population for restricted delivery or counteract pathogens that release chemokine-binding proteins to subvert human immune responses [20].

## 4. Conclusions

In the present manuscript, we proved that chemokine-decorated NPs are versatile nano-tools for selective delivery applications. It is possible to use different chemokines to reach diverse cell targets or the same cell in different stages of maturation and expression of cognate receptors. Pathological states often change cell surface protein expression as well, and chemokine-decorated NPs could provide a novel future solution to precisely drive drugs or vaccine cargos.

## Figures and Tables

**Figure 1 nanomaterials-12-03560-f001:**
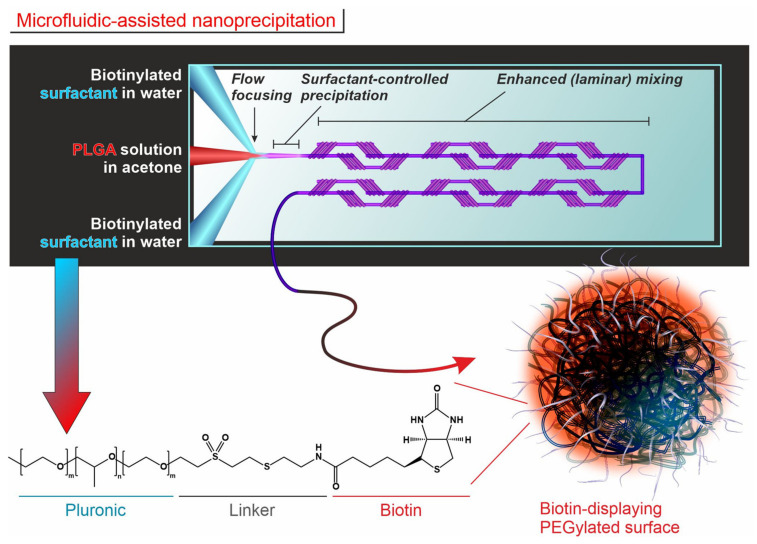
Scheme of the process of PLGA nanoprecipitation in the presence of functional surfactants (biotinylated Pluronic) in a microfluidic set-up. A polymer organic solution (PLGA in acetone) was mixed with an aqueous solution of a surfactant in a passive mixer chip; the resulting precipitation of the polymer occurred after a flow-focusing step that *inter alia* avoided the accumulation of material on the channel walls and was limited to a sub-micron size by the in situ presence of the surfactant. The extraction of the organic solvent from the nanoprecipitate was ensured by the successive efficient laminar mixing in the chip. The chemical structure showed the triblock polymer structure of Pluronic (on the left) and its functionalization with biotin (only on the right for simplicity, but the reader should understand that Pluronic was derivatized on both ends). At the end of the process, the PLGA surface was covered by surfactants entrapped through their hydrophobic blocks in the PLGA matrix.

**Figure 2 nanomaterials-12-03560-f002:**
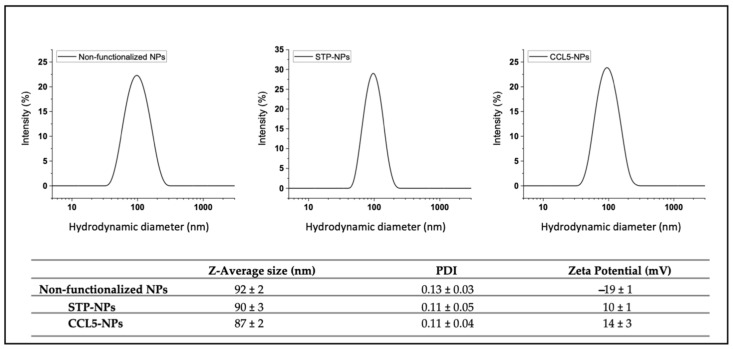
Nanoparticle characterization. DLS data of non-functionalized NPs, streptavidin- conjugated NPs and CCL5-functionalized NPs. Every plot shows NP diameter (nm) on X-axis and intensity (percent) on Y-axis. Table shows Z average size, PDI and Z potential.

**Figure 3 nanomaterials-12-03560-f003:**
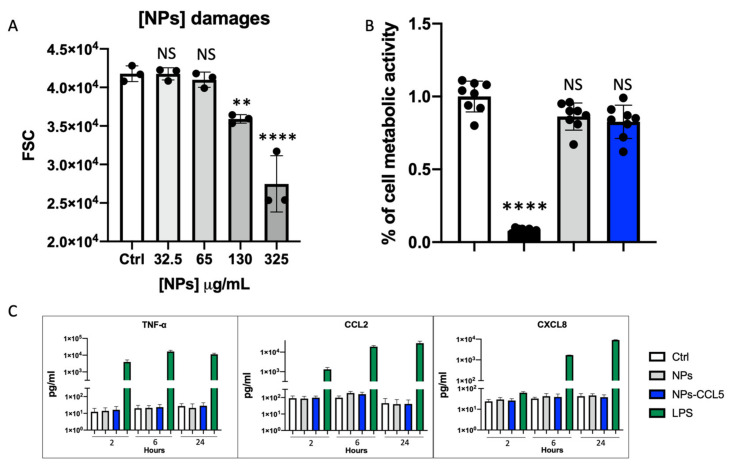
Nanoparticle biocompatibility. (**A**) Flow cytometry histogram represents THP-1 cells treated with different concentration of streptavidin-conjugated NPs ranging from 32.5 to 325 µg/mL. FSC parameter on *Y*-axis displays the relative cell size. Significant differences were determined by ANOVA multiple comparison vs. control column, NS *p* > 0.1, ** *p* < 0.01, **** *p* < 0.0001. (**B**) MTS measured metabolic activity of THP-1 in 5% HS supplemented medium. Data are represented in percentages relative to untreated control. Untreated THP-1 cells (Ctrl, white bar), THP-1 cells treated with 65 µg/mL streptavidin-conjugated NPs (gray bar) and 65 µg/mL CCL5 NPs (blue bar). As positive control, 10% DMSO (black bar) was used. Significant differences were determined by ANOVA multiple comparison vs. control column, NS *p* > 0.1 **** *p* < 0.01. (**C**) NP-induced cytokine release. Column graphs show the concentrations of TNF-α, CCL2 and CXCL8 released by cultured THP-1 cells after treatment with 65 µg/mL streptavidin-conjugated NPs (NPs), CCL5 NPs or 100 ng/mL LPS for 2, 6 and 24 h measured by Bio-Plex MAGPIX Multiplex Reader. Control (Ctrl) represents untreated cells. Results are expressed in pg/mL. All bars show the average of at least three independent experiments +/− SD (error bars). No significant differences were observed in the different NP samples by ANOVA multiple comparison vs. control column, NS *p* > 0.1.

**Figure 4 nanomaterials-12-03560-f004:**
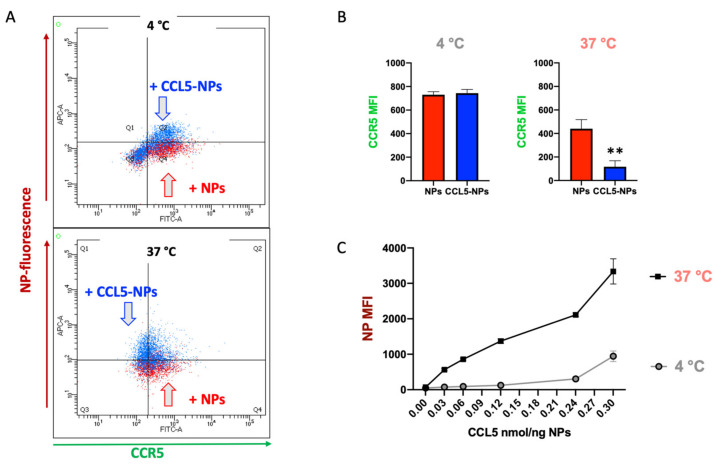
(**A**) CCL5-NP-modulated CCR5 expression at 4 °C and 37 °C. Flow cytometry dot plots showing CCR5 expression onto THP-1 membrane (logarithmic scale, X-axis) in the presence of CCL5 NPs (blue dots) or control NPs (NPs, red dots). (**B**) CCR5 expression at 4 °C and 37 °C in the presence of NPs and CCL NPs. Significant differences were determined by unpaired *t*-test, ** *p* < 0.01. (**C**) Increased binding (4 °C) and internalization (37 °C) of CCL5 NPs dependency on the amount of biotin-CCL5 bound onto NP surface.

**Figure 5 nanomaterials-12-03560-f005:**
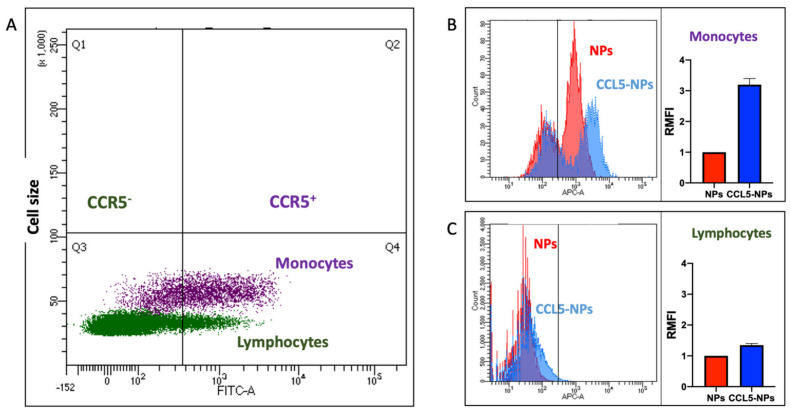
(**A**) CCR5 expression on primary monocytes and lymphocytes. One out of three different PBMCs donors is shown as representative sample. Highly CCR5-expressing monocytes (purple dots) and low CCR5-expressing lymphocytes (green dots) are shown. CCR5 antibody staining in logarithm X-axis. Relative cell size in FSC (linear Y-axis scale). Selective CCL5 NP internalization in CCR5^+^ monocytes (**B**) and lymphocytes (**C**). CCL5 NPs (blue histogram and column) vs. streptavidin-conjugated NPs (NPs, red histogram and column) treatment at 37 °C for 45 min in 5% HS complete medium. Columns show triplicates of the same PBMC preparation.

## Data Availability

All data reported are provided in the main text and in the supporting information.

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
