# Peer review of "Chemokine-Decorated Nanoparticles Target Specific Subpopulations of Primary Blood Mononuclear Leukocytes"

_nanomaterials, 2022, doi:10.3390/nano12203560_

Round 1

Reviewer 1 Report

In this Manuscript by Anissa Pisani et colleagues, the authors showed that pro-inflammatory CCL5 decorated-NPs can discriminate among primary mixed leukocyte sub-populations differently expressing CCR5.

The topic is interesting, the paper is well written and organized. However, there is still some room for improvement and I have the following comments for that.

The introduction section is too short, add more data on decorated nanoparticles and biological recognition to improve treatment and diagnosis.

A summarized scheme with all steps of the study would be welcome to attract readers.

The Authors should read the article for typos and stylistic errors that appeared in the article.

What is the novelty of this paper?

What perspectives for human health does this MS have?

Consider revision accordingly.

Author Response

In this Manuscript by Anissa Pisani et colleagues, the authors showed that pro-inflammatory CCL5 decorated-NPs can discriminate among primary mixed leukocyte sub-populations differently expressing CCR5.

The topic is interesting, the paper is well written and organized. However, there is still some room for improvement and I have the following comments for that.

- The authors thank the Reviewer for her/his comments and suggestions.

  1. The introduction section is too short, add more data on decorated nanoparticles and biological recognition to improve treatment and diagnosis.

- As advised by the reviewer, the introduction and all figures have been fully revised. Recent biomedical works on chemokine receptors targeted with nanoparticles modified by small-peptide or small-antagonist molecules has been cited.

  1. A summarized scheme with all steps of the study would be welcome to attract readers.

- We introduced a scheme of the project in the Supporting Information to reduce the number of figures in the Brief Report format.

  1. The Authors should read the article for typos and stylistic errors that appeared in the article.

- The paper has been revised and some sentences rephrased.

  1. What is the novelty of this paper?

- Coating polymer nanoparticles with an entire inflammatory chemokine, and not with small-antagonist molecules or modified-small peptides, to target specific cell subsets expressing its cognate chemokine receptors is a novel approach. The discrimination of the target cell in a mixed population proves the reliability of, in our opinion, a valuable “Trojan Horse” using inflammatory mediators to target cells activated by such mediators. Furthermore, the exploitation of streptavidin-biotin binding to coat the particle with several biotinylated-chemokines (or other cytokines) make the method very easy to be tailored for different targets in different physiological or pathological conditions.

    5. What perspectives for human health does this MS have?

- The possibility to tailor NP-carriers for specific targeting of cell mediating inflammatory reactions could lead to personalized therapies. Applying future therapies by means of diverse chemokine receptors’ expression in the different cell populations and in different pathological conditions, could reduce the off-targets and maybe the dose of the drug.

Reviewer 2 Report

In the manuscript titled "Chemokine decorated nanoparticles target specific subpopulations of primary blood mononuclear leukocytes", Pisani et al. decorated the surface of biocompatible polymer nanoparticles with full-length CCL5, an inflammatory chemokine attracting leukocytes by binding CCR5 highly expressed in the blood circulating monocytes. The CCL5-NPs delivered to PBMCs were selectively internalized by CCR5+ monocytes but not by CCR5- lymphocytes, which shows that pro-inflammatory CCL5 decorated-NPs can discriminate among primary mixed leukocyte sub-populations differently expressing CCR5, proving that chemokine-decorated NPs are versatile nano-tools for selective delivery applications. This work is well designed, well developed and supported by the results. The authors considered the effect of the protein corona, which has easily been underestimated by others. The authors also did an adequate analysis of nanoparticle biocompatibility, which is important for targeting leukocytes. After addressing the following questions properly, the manuscript could be considered for publication.

1. The NPs were prepared in a microfluidic-assisted nanoprecipitation method, and it will be better for readers’ understanding if the authors can add a scheme or photo of the microfluidic device.

2. L132: “The use of a biotinylated CCL5 chemokine did not dramatically altered the average NP size, nor the 133 breadth of its distribution (thereby discounting aggregation), nor surface charge.”

The “altered” here should be “alter”. The surface charge change from -19 to 14 mV should be considered a significant change. Please revise and discuss it.

3. The authors should analyze the statistical differences between different groups for all the figures.

4. The introduction should contain more dated literature, especially research articles. If no one else has been working on that topic in recent years, one may wonder if the impact of that research is actually high.

Author Response

In the manuscript titled "Chemokine decorated nanoparticles target specific subpopulations of primary blood mononuclear leukocytes", Pisani et al. decorated the surface of biocompatible polymer nanoparticles with full-length CCL5, an inflammatory chemokine attracting leukocytes by binding CCR5 highly expressed in the blood circulating monocytes. The CCL5-NPs delivered to PBMCs were selectively internalized by CCR5+ monocytes but not by CCR5- lymphocytes, which shows that pro-inflammatory CCL5 decorated-NPs can discriminate among primary mixed leukocyte sub-populations differently expressing CCR5, proving that chemokine-decorated NPs are versatile nano-tools for selective delivery applications. This work is well designed, well developed and supported by the results. The authors considered the effect of the protein corona, which has easily been underestimated by others. The authors also did an adequate analysis of nanoparticle biocompatibility, which is important for targeting leukocytes. After addressing the following questions properly, the manuscript could be considered for publication.

- The authors thank the Reviewer for her/his comments and suggestions.

  1. The NPs were prepared in a microfluidic-assisted nanoprecipitation method, and it will be better for readers’ understanding if the authors can add a scheme or photo of the microfluidic device.

- As suggested by the Reviewer, we introduced a scheme of the microfluidic-assisted nanoprecipitation and chemical reactions as Figure 1.

  1. L132: “The use of a biotinylated CCL5 chemokine did not dramatically altered the average NP size, nor the 133 breadth of its distribution (thereby discounting aggregation), nor surface charge.” The “altered” here should be “alter”. The surface charge change from -19 to 14 mV should be considered a significant change. Please revise and discuss it.

- According with the reviewer, we modified the text (“atered” to “alter”). We agree with the reviewer that STP modification from – 19 mV to + 14 mV is a significant NP zeta potential change as already mentioned in L130-L131. The sentence in L 135 was referred to biotin-CCL5 vs STP decoration of NPs. We clarified it adding a further sentence.

  1. The authors should analyze the statistical differences between different groups for all the figures.

- Statistical differences between the groups of NPs vs CCL5-NPs using several replicates of THP-1 cells have been analyzed by AVOVA multiple comparison analysis vs. control columns followed by Dunnett post hoc test. ** p < 0.01, ***p < 0.001, **** p < 0.0001, or unpaired t test with Welch’s correction where indicated. Figure 4 columns (on the right) show the average of 3 replicates of one representative sample out three different PBMCs donors. Although the results’ trend is very similar in the experiments with the 3 different cell preparations, we did not statistically merge the 3 donors’ PMBCs results as CCR5 expression on monocytes is slightly different in each donor.

  1. The introduction should contain more dated literature, especially research articles. If no one else has been working on that topic in recent years, one may wonder if the impact of that research is actually high.

- We thank the Reviewer for the improving suggestion. Introduction has been fully revised. Very recent works (2022) on targeting CXCR4 with nanoparticles modified by small-peptide or small-antagonist molecules has been cited and discussed.

Round 2

Reviewer 1 Report

No answer given.